# It’s Not Just Me, It’s Us, Together: The Embodied of the Wounded Healer in the Role of Sex Trade Survival Mentors—A Critical Mentoring Perspective

**DOI:** 10.3390/ijerph20054089

**Published:** 2023-02-24

**Authors:** Dodish-Adi Kali, Menny Malka

**Affiliations:** The Spitzer Department of Social Work, Ben-Gurion University of the Negev, Beer-Sheva 84105, Israel

**Keywords:** critical mentoring, feminist research, rehabilitation, women in sex trade, wounded healer

## Abstract

Mentoring is one of the unique forms of rehabilitation used to engage with women in the sex trade. The role creates personal and professional challenges; one concerns the mentors themselves dealing with a past in the sex trade, embodying within it a sign of social disgrace. Echoing the concept of the “wounded healer,” the present study examines how mentors who are sex trade survivors perceive their role in supporting the rehabilitation of women in the sex trade and the meanings that they give it. The research is based on a qualitative approach from a critical-feminist point of view. Eight female mentors and sex trade survivors, working in different settings, participated in the study. Data collection was conducted through semi-structured, in-depth interviews. Based on content analysis, the study points to four components of mentoring vis-à-vis the rehabilitation of women in the sex trade: (1) mutual identification and shared destiny; (2) corrective experience; (3) hope; and (4) saving lives. In addition, mentoring serves as a bridge for the mentors, eliciting opportunities for growth transforming out of their pain. The research findings are discussed in the context of the theoretical framework of critical mentoring, and how a relationship and a therapeutic alliance can turn mentoring into a critical healing practice, in relation to four principles: (1) equality; (2) critical empathy; (3) recognition; and (4) solidarity. The paper encourages the use of mentoring-based interventions in the process of rehabilitating women in the sex trade.

## 1. Introduction

Mentoring describes a supportive relationship between two people, one of whom has a greater degree of experience and knowledge in a specific area or field of knowledge, facilitating the transfer of knowledge and a process of empowerment through the engagement between the mentor and the mentee. In the “helping relationship” professions, there is an ongoing debate regarding the encounter between the history of the mentor and that of the mentee, mediated by the concept of the wounded healer. Over the years, in the spirit of the critical-feminist approach, the practice of mentoring has been used with women in general, and especially women from vulnerable groups, under the rubric of critical mentoring. The objective of the research presented in this article was to examine the mentoring of women in the sex trade (we would like to point out that the research participants (women in the sex trade, WST, who became sex trade survivor mentors, STSM) constitute one of a number of groups that are included in the definition of sex trafficking but do not form a part of the current study. The authors are aware of the existence of different definitions; in this specific study we use the definitions of WST and STSM) (WST) by sex trade survivor mentors (STSM) under the conceptual frameworks of critical mentoring and the wounded healer.

In the field of research and practice regarding WST, it is commonly acknowledged that WST constitute an extremely vulnerable and victimized group. This manifests in the lack of control they have over their lives; it also contributes to the social and cultural barriers they encounter due to stigmatization and the recurring impact of trauma from childhood and accumulated during life—all this with the overhanging presence of sexual, physical, and mental assault [1,2]. WST suffer from physical health problems, high levels of suicidality, anxiety, and depression, and in addition must cope with post-traumatic disorders and addiction problems [3,4].

These deep mental and physical vulnerabilities, along with the stigma of socio-cultural exclusion, pose a challenge to aspects of rehabilitation work with WST, including their capacity to access health care services. These factors also erect barriers hindering the creation of trusting relationships between WST and the welfare professionals seeking to work with them and on their behalf [5,6]. There are clear indications that engaging in the sex trade (ST), and the traumas associated with it, invite the use of mechanisms of dissociation, self-appropriation, and false selves—phenomena that respond to trauma-informed psychotherapeutic intervention [7]. However, there are indications that psychotherapy as a central intervention is not sufficiently accessible to WST; even when psychotherapeutic interventions are available, it is unclear how successful and effective these are with respect to WST [8]. However, clinical evidence based on a single case study points to the importance of the therapeutic alliance and of the relationship between the therapist and the patient as a central component of trauma-informed care for WST—a quality of care based on bonding and a trusting relationship, thus creating a safe place for the WST survivor [9].

Regarding the vulnerability and social exclusion of WST, it is clear that rehabilitation processes should be based not only on specific therapeutic interventions focusing on mental trauma, but also on psychosocial interventions, and that the complementary approaches coordinated by dedicated aid organizations should operate in line with principles of systemic intervention. One unique form of intervention, which stems mainly from the principles of alliance and the therapeutic relationship—that is, a relationship based on trust, honesty, acceptance and a non-judgmental approach on the part of the therapist, and designed to serve the patient’s change processes—is an intervention based on peer support in the form of mentoring, in which WST are given emotional support, acceptance, and hope [10,11]. A small number of studies exploring how WST experience peer-support mentoring have shown that this form of intervention provides them with the motivation to engage fully with rehabilitation processes [12,13]; engenders the development of resistance to the stigmatization they face [12]; creates a safe place to share; allows for the normalization of negative feelings in the face of past events; and gives them a role model [14]. Thus, the position and role of a mentor links to the importance of the therapeutic alliance and the helping relationship; the mentors have “been there” (the research described in this article discusses a specific group of mentors with similarities between them and their mentees, their link deriving from the fact that the mentors have faced similar challenges in the past. Thus, this specific group does not reflect other configurations of mentoring in which there are no linkages of this kind between the mentor and the mentee) and are now helping others in a similar situation. This is a unique position that aligns with the concept of “the wounded healer” [15,16]—that is, the idea that therapists have experienced deprivation and psychological wounds in their past, which may influence the way they experience their patients and the communication between them [17].

There are, however, relatively few studies that have systematically examined how STSMs describe and give meaning to their role as mentors of WST. For this reason, the focus of this study is on STSMs as a unique group not usually accessible to researchers [14]. Thus, taking a critical perspective, the objective of this research is (1) to examine the contribution of STSM and their knowledge to understanding the experiences of WST; and (2) to examine the meanings of mentoring that are informed by past experiences, i.e., the wounded healer, for the WST rehabilitation process.

### 1.1. The Wounded Healer

In this study, conceptualization of the notion of the wounded healer derives from the understanding, which has evolved over time, that people who have successfully dealt with specific challenges (for example, addictions or mental health challenges) may possess a unique capacity to help people dealing with a similar problem, and that both parties may benefit from this [15]. The conceptual origin of the wounded healer comes from Greek mythology and the proposition that a subject can draw from the experience of their own wound to help others [16]. Thus, as has been shown by various studies, the quality of the wounded healer position is reflected in their ability to create an empathetic relationship with people receiving assistance from them—a relationship anchored by compassion, direction, and support, which can deepen the engagement between them and can ultimately support post-traumatic growth [17].

However, the wound to which the concept refers is located across a wide range of possibilities, serving as a basis for the empathic bridge built on it between the giver of help and the recipient of help [16]. In general, this continuum is derived from the body of knowledge about the wounded healer. This can be divided into two main frames of reference.

The first originates in the field of professional helping relationships in various areas of therapeutic support, such as counselling, social work, psychological intervention, and therapy. As part of their professional training, practitioners are trained to make intelligent and controlled use of these wounds and painful personal experiences in building an empathic bridge toward their patients [17].

In contrast, the second body of knowledge relates to volunteers or colleagues with experience of similar challenges, or indeed the same challenge, as the people they are working with, as in the case of ex-prisoners [18], addicts [19], mental health service users [20], and women on the margins and in situations of risk [21]. One expression of this, and one of the most common and relevant forms of semi-professional help relationships, is the practice of mentoring. Mentoring provides a frame of reference for this study and for WST as the studied population group.

### 1.2. Critical Mentoring

Since the 1960s, there has been steady growth in the use of mentoring as a semi-professional practice [17,18,19,20]. Mentoring, in its traditional definition, refers to a long-term teaching and learning relationship in which the knowledge and skills of the mentor are transferred to the mentee, who is usually younger and less experienced than the mentor [22]. Thus, in cases where the mentor has experience or a history similar to that of the mentee, mentoring is linked to the conceptualization of the wounded healer.

The critical field of mentoring has developed over the last decades, framing the mentor’s relationship with the mentee as a potential space for activism, for resistance to inequality and various forms of oppression, and for the promotion of social justice [23,24]. Critical mentoring conceptualizes the mentoring process as a space for facilitating processes which extend beyond the acquisition of social skills and competencies. To this end, the principles anchoring the mentoring process include the reconstruction and co-construction of knowledge about the world; developing the ability to reflexively observe the self, different identities, and social positions; building partnerships and relationships while reducing and resisting power relations; and the acquisition of different types of knowledge and capital as a means of resisting various forms of inequality [25].

Assimilation of the ideas of critical mentoring into the field of mentoring is based on two complementary principles. The first emphasizes the effort required to expand the inclusion of marginalized populations, and thus a point of view and knowledge based on experience, practice, and research about mentoring, by way of participatory and qualitative research methods [26,27]. The second principle is based on the use of critical theories as an interpretive framework for analyzing, conceptualizing, and giving meaning to the various processes that occur within the mentoring framework. These goals are anchored by concepts such as power relations, unequal social structures, and the pursuit of social justice, together with the desire for transformative change and the need to develop more empowering and democratic practices of mentoring [28].

Drawing from the superstructure of critical theory opens possibilities for considering within the mentoring framework not only the psychological characteristics of the mentee, but also aspects of the broader reality within which barriers and challenges emerge, such as barriers emanating from the social categories of race, ethnicity, class, gender, and the intersection of some or all these categories [27,29]. For example, adopting a feminist critical point of view to mentoring emphasizes the counterweight that mentoring can provide to more traditional approaches that may preserve gender power relations [30,31]. The uniqueness of the feminist perspective is reflected in the creation of a mentoring relationship based on partnership and reciprocity, and in the integration of democratic values in the pedagogical-educational process; while the task-focused technocratic aspect is reduced, the procedural aspect and the feminist elements of gender identity awareness are strengthened [24,29,30,32].

Regarding the structural barriers that may confront a mentee, critical mentoring takes into consideration the systemic and structural barriers experienced by minorities and marginalized individuals and communities. Barrier transcendence theory [33] proposes that mentoring brings into consideration, alongside the characteristics of the individuals concerned and the events of their lives, the factors that create motivation for change, as well as the barriers and opportunities that exist within the environment and the given political and historical context [34].

Thus, in the field of critical mentoring research and practice, barrier-overcoming theory serves as a complementary theory to feminist theory in that it emphasizes the activist role of the mentor. The mentor sets out to critically analyze social reality and to identify barriers in the social-cultural-class integration of the vulnerable and marginalized populations she mentors. On this basis, the mentor initiates actions of active advocacy to remove barriers originating in various forms of inequality, exclusion, and the lack of social and cultural capital. The mentor perceives mentoring as a space for correcting social injustices [33,34].

Another component of critical mentoring is based on the use of academic research to enable the voices of vulnerable and marginalized populations to be heard [31]. This move is based on the perception of WST and STSMs as possessing unique knowledge which is not expressed. Thus, the absence of the STSM point of view in the research and practice of mentoring emphasizes the feminist-critical principle, which encourages giving voice to the knowledge and lived experience of STSMs. That is, the point of view of women who went through a process of empowerment and subsequently became mentors to WST, and who can observe the experience from different perspectives—before (as WST) and after becoming mentors [14,31]—is centered.

### 1.3. Research Questions

The aim of this study is to examine mentoring relationships between WST and STSMs, under the conceptual frameworks of critical mentoring and the wounded healer, with respect to three questions:(1)How does mentoring contribute to the WST rehabilitation process from the STSM point of view?(2)How are the past experiences of the STSMs (the wounded healers) embodied in their work with WST?(3)What do STSMs gain from being mentors?

## 2. Method

The research was conducted using a qualitative-phenomenological approach. This approach seeks to understand social phenomena in a holistic manner based on fundamental examination of the meanings that such phenomena are given by research participants [35]. In the current study, a feminist critical perspective [36] was integrated into this approach to link the knowledge of the participants to their gendered social contexts—in the main characterized by exclusion, inequality, and oppression. The study perceived STSMs as fulfilling a semi-professional role lacking a defined social status and as people whose voices may be silenced.

### 2.1. Participants

Eight STSMs participated in this study. This is a relatively small sample, albeit consistent with the relatively small scope of STSMs in Israel. The sample size also reflects the challenges in research involving vulnerable populations in general, and WST in particular [14,37]. Two of the STSMs were (at the time of the study) single; one was a widow, and the other five were married. Four were mothers. Their length of experience as mentors for WST ranged from 5 to 15 years. The number of years that had passed since the participants had left the ST ranged from 9 to 25 years.

The participants in the study had all served as mentors in support organizations and dedicated frameworks for WST. For the most part, their training as STSMs is considered another step in the rehabilitation process that they went through in their attempt to free themselves from their involvement in the ST. For some of them, this rehabilitation process was not linear, but was instead characterized by ups and downs because they faced various addictions that had a direct connection to their involvement in the ST. As part of their duties, STSMs were required to carry out activities of reaching out, locating, and recruiting WST in the street, building relationships of trust with them, and serving as a resource in times of need. They were also members of multi-professional teams, held formal and informal conversations with WST in rehabilitation processes, and sometimes served as mediators between the WST and professionals. As part of their duties, STSMs received training and guidance from the social work professional in the organization where they worked.

Thus, recruitment of the participants to the study was on a voluntary basis; they did not receive compensation and were told that the study serves as an opportunity to contribute to knowledge creation about mentoring with WST.

### 2.2. Procedure

Participants were recruited in three stages. First, an appeal was made to social workers working as contact persons in aid organizations working in the field of rehabilitating WST, and who employed STSMs. The aims of the study were described to the social worker in telephone conversations along with consultations regarding the recruitment procedure of the participants. In the second stage, contact was made with three participants who had expressed a willingness to participate in the study. In the third stage, the snowball recruitment method was used to expand the number of participants [38] through social and professional networks and with the assistance of the first three interviewees. Adhering to the principles of feminist research [36], care was taken that the place of the interview was determined according to the request of the women—generally, on the premises of the organization where they worked or in other places that were convenient for them, ensuring that participants were given the opportunity to suggest a location that suited them.

### 2.3. Research Tool

The data were collected through semi-structured, in-depth interviews [39] conducted by the first author. Each interview opened with the invitation: “Tell me your story as a STSM” and continued based on an interview guide including questions exploring the role of the STSM from the participants’ perspectives. For example: “What does being a mentor mean to you? Tell me about a moment that represents what it is to be a mentor? What is the impact of being a mentor on your life? Tell me about challenges and difficulties in the role? How do you deal with them?” The interviews were conducted according to principles of critical-feminist research; these include personal disclosure, reciprocity, avoiding intrusive questions, and giving room for questions and choice. The goal of this approach was to give space to the knowledge and expertise of the participants and to reduce perceptions of a hierarchal distinction between interviewee and interviewer [40].

### 2.4. Data Analysis

The interviews were recorded, transcribed, and then analyzed using categorical content analysis [41] across a number of stages: (1) A holistic reading of the interviews and initial identification of meanings; (2) division of the interviews into segments identified as units of meaning (for example: emphasizing the similarity over the differences in mentor-mentee relationships; love; caring relationships; caring; respect); (3) grouping said units of meaning into central themes relevant to the research questions; (4) a fresh holistic reading to confirm that the logical connections developed between the various topics correctly reflected the raw data. The study’s reliability was verified by way of triangulation [42]: (1) a comparison of the researchers’ data analyses was conducted; (2) the study’s results were compared against the relevant literature on WST and critical mentoring; (3) the research results were presented at three different conferences; (a) at a dedicated conference on practice and research with WST; (b) in a designated session for practice and research with WST as part of a social policy conference; and (c) a conference on qualitative research. This combination made it possible to present the research results to researchers in the field of WST, professionals and experts in the field of WST, and to qualitative researchers. The feedback contributed to refining the significance of being in the WST story as part of the STSM life story, and the connection between WST and STSMs through the concept of the wounded healer.

### 2.5. Ethical Considerations

The research plan was approved by the ethics committee of the Social Work Department at Ben-Gurion University, Israel. Participants all signed an informed consent form; personal details were retracted to ensure confidentiality and to preserve their anonymity. The participants were informed that they were free to withdraw their consent during or after the interview, or before the publication of the data in the research report. In addition, considering that they are part of a unique population group within the Israeli context, when presenting the data we redacted and/or changed details that could reveal the identity of the participants (such as ages, place of work, and related status). Thus, when presenting the sample, we used generalized data without discussing specific data relating to any one participant. To maintain anonymity, pseudonyms are used for all the participants.

In addition, the interviews (conducted by the first author) were conducted sensitively to avoid putting pressure on the interviewees and to give them a sense of control over the process. During the preliminary phone call, interviewees were told that they could withdraw their consent to participate in the interview at any stage. During the interviews, it was made clear to the participants that they are not obligated to answer any question and that they could stop the interview at any time. At the end of the interview, the interviewer checked with the interviewees whether the interview had been overly challenging and gave them her phone number so that they could report to her about any distress they had experienced as a result. The interviewer contacted each interviewee a few days after the interview to make sure they were not experiencing secondary emotional trauma due to the interview. None of the interviewees reported experiencing distress following the interview, or later.

## 3. Results

The results of the study point to five main themes providing information about the way in which the notion of the wounded healer is embodied within the STSM role. While the first four themes refer to the STSMs’ perceptions of the contribution of mentoring to the process of rehabilitating WST, the fifth theme points to how, from their point of view, mentoring contributes to the continuing process of STSM recovery, offering a platform for personal-professional growth.

### 3.1. Mutual Identification and Shared Destiny

The interviewees underlined mutual identification and shared destiny as a formative principle in establishing their relationship with WST and in the way that they bonded with them, whether on the street or in the institution responsible for the rehabilitation process.

**Talia**: Essentially me and her are the same… I told her the difference between you, and I is that I have some tools to deal with (the consequences of ST) but I feel the same feelings as you—which are difficult and not easy, and make you feel like kicking everything and saying, “Damn it”.

Talia points to emphasizing similarity over difference as a way of creating reciprocity. She makes sure to point out the difference between her and the WST with whom she meets, but only to make this distinction unnecessary, all the while emphasizing the imagery through the emotional component and feelings of rage towards the world. Emily took a similar line, this time regarding the connection between addiction and drug withdrawal:

**Emily:** There isn’t much difference between us. After all, there is only one small thing that differentiates me from her when she uses the drug, which is actually the stripe of the substance… and if I don’t do my way right, then the day after tomorrow I could [find myself] back on that side again.

Emily also emphasizes the greatness of the imagination over difference through the element of addiction. As a STSM who is also a “clean addict”, and through her understanding that many WST have addiction struggles, she used the symbolic term of a “thin drug strip” to highlight the thin difference between her and a WST. Later in the interview, Emily pointed to humility as another principle of mutual identification and shared destiny:

**Emily**: I hug a lot, for example, when I meet them on the street, so as not to let them feel, even for one moment, that there is a difference between us... I constantly find this point in me when I am at my most humble. Otherwise, I don’t approach them, because I don’t want to put myself in a position of arrogance, which would provoke the feeling or thought that I am different or better than them.

Mutual identification and shared destiny, in this case, is reflected in Emily’s willingness to embrace her mentees; she carefully considers how she acts with and toward the WST she meets in the outreach process on the street so as not to highlight hierarchical relationships. From this, one can learn the importance of humility as a human and semi-professional position at the same time—especially in Emily’s work as a mentor with women stigmatized due to associated with the ST. Talia also reinforced the idea that one should be careful of arrogance:

**Talia**: Whoever says that you don’t need those you take care of, [it] is arrogance, it’s a lie. I need them, I also tell them, I need you as much as you need me—the same thing one-on-one, the same thing.

There is an alliance embodied in the relations of mutual dependence. Talia describes how she reveals to the WST that she needs them as much as they need her to continue her path; she sees herself through them in a way that adds meaning to her role as a mentor. Masha referred to her manner of dress when in the role of mentor, linking this to mutual identity and shared destiny:

**Masha**: It’s not pleasant when I come [to the street] well dressed, all tidy… I prefer to wear the simplest shirt and pants, and to be with her and let her feel that I’m with her—that there should be no gap, that there should be no distance [between us] … However, clothing makes a statement mainly for women who are survivors, so I prefer to adjust myself.

Mutual identity and shared destiny are woven into Masha’s approach, an understanding drawn from her own experience as a sex worker. This manifests in the form of a sensitive process of adapting her clothing to the situation on the street so as to avoid suggesting, even inadvertently, a hierarchy between them, in the understanding that this can be a sensitive issue in a street encounter with WST.

### 3.2. Curative Experience: Being in the Position of a Parent

Another element of the role of STSM manifests in the form of the parental-motherly position that they take as a way of facilitating curative experiences:

**Talia**: In my work at the hostel, I felt that even though they were women, sometimes even my age, oh no, older than me too […] and yes, I felt that I was a model of a mother for them.

Here, Talia describes being in a parental-motherly position as a model suitable for engaging with WST survivors—even if older than her. This position is translated into actions, forms of reference, and the understandings that the STSM points out:

**Masha**: I will make Blinchas [a type of pancake] for her in the middle of the night. She is hungry and she wants [to eat] even though the hostel doesn’t prepare food at night, but it’s like she’s so lonely and she’s been neglected for several years, and she hadn’t had it [to eat], and she has an eating disorder, and she doesn’t eat and suddenly she eats and doesn’t throw up, that also means something and, that’s my motherly part coming out.

The ability to see the mentee where she is is reflected in the flexibility of the shelter rules and the willingness to indulge her by preparing “special” food for her. Masha describes how this move, which takes on meaning as a “maternal” move in its essence, helps to establish trust with the mentee, contributing to her capacity to address her eating disorders.

This maternal role is reflected in events on the street:

**Anat**: I received beatings from sex consumers who came to look for the girls, and from the police with search warrants. It was at the very beginning that everything was hacked and there were no clear rules and I was protecting the girls… No one will enter, [over] my dead body, and no one will do anything to them, [it is] as if they were one of my children.

Anat calls the WST whom she supports “girls” in a way that embodies a mother’s way of relating to her children; in the passage quoted above, she describes how she mobilizes to protect them with her body, like a lioness protecting her cubs.

Added to this is the ability to give warmth and love:

**Emily**: Hugging a sex trade survivor and saying “I love you” and that she is able to say the same words back, apparently it doesn’t sound like much... but love is an “issue” for a person who doesn’t like to say these words or accept or open one’s heart in general and to believe that someone can love him or her… for a person who in the midst of self-destruction, where there is no love, there is no…there is death as if there is death within life, oh and a desire not to be… and hate big, huge, first of all [directed] towards myself... to give a hug and receive a hug and say I love you… to accept… or believe it, or maybe try to believe that the mentor or social worker says it from the heart—it’s an achievement, it’s truly a miracle.

The passage above reveals a basic and deep understanding of how the severe traumas and vulnerabilities experienced by WST throughout their lives are translated into self-hatred and a feeling that they are not worthy of love. This is the place where a loving and compassionate parental-motherly position can be translated into a hug, into a meaningful human touch. According to Emily’s words, this is an approach that is sensitive to the hardships of the lives of ST survivors—not a suffocating or stifling hug.

### 3.3. Hope: Holding the Belief That Rehabilitation Is Possible

The mentors point to hope as a third main component in their role in supporting WST:

**Emily**: The fact that they are addicts and on the street, and that I got rid of drugs and made a way with myself makes a difference between us that can give…give them hope only in this sense is a difference not beyond that. It can [free them to] strive for something else, or to show them some kind of way when they are blocked or do not believe that they can do it.

In most cases, attempts to blur the differences between mentor and mentee were emphasized so as not to create a feeling of condescension toward WST. That said, it is important to emphasize a specific difference: the mentor has been successfully rehabilitated, compared to WST who are still on the street. However, this emphasis is intended to create hope and the belief that rehabilitation is feasible. Sometimes, the element of hope is embodied in the STSM’s meeting point with their past as WST:

**Talia:** I think it’s the most tangible thing for them to see that I was with them [in the prison] as a prisoner, and here’s a fact it’s possible to get out of it… for them it was very fundamental and very strong… this is my mission… I pass on the message of hope.

This situation where women from the past see how the mentor, with a shared past history—in prison and/or on the street—has recovered provides the basis for creating meaning and a sense of mission, embodied in the role of the mentor. The mentors understand that the dream of recovery is a dream that every WST holds deep inside her:

**Masha**: I think yes, that every woman who knows me and knows that I was both in prison and in prostitution, I’m sure she does also want to be in this place. She doesn’t know how, maybe she doesn’t like how it can be done, but I think that every woman has a desire to get out of where she is when she comes home and puts her head on the pillow, she doesn’t want to be where she is.

Indeed, the presence of the mentoring, and the fact that she is a WST survivor, provides a trigger for the dream and hope for change buried deep within the soul. In this way, mentoring can work by providing a role model—a role to which one can aspire. Alma encapsulated this process in a metaphor of light versus darkness:

**Alma**: For each of the women, the light needs to be turned on for her, this place of hope and faith… You know, part of the work is also making quarterly and semi-annual reports on how many women get out of prostitution and the like… I don’t like it, I write it, I follow it and that, but for me every woman is a world, and I think my vocation is to bring lots and lots of light, love, and hope.

Along with the administrative aspects of the mentoring work, it is the sense of mission, the knowledge that they are proof of the possibility of leaving the darkness and entering the light; this is what charges them with the energy and passion that they invest in their work.

### 3.4. Saving Lives: This Is What Motivates Us

Another element of the mentor’s work is embodied in the inherent and continuous danger that external circumstances pose to the lives of WST:

**Rita**: Maybe I will really save one person? It is a fact that my mentor, that when she gave me her phone number, she saved me… We (the mentors) talk to each other in the hostel, [saying to ourselves] “I wish we could have saved them all.” But our percentages are something like one in ten.

The accessibility of Rita’s mentor when she was in the ST helped nurture a deep understanding that something as seemingly banal as lending a hand actually saves lives. Even though the successful rehabilitation rate is low, the desire to act to help as many WST as possible is an essential component of the mentor’s work:

**Rita**: I understand how much meaning this meeting has between a woman who needs help and a woman who has been in this place that she just understands and that she can stay there in the dark with her and hold her hand and not run away from her... it really saves lives… I think this is the essence of me my role actually and this, this is the thing that really motivates me today the most.

Rita ties the element of rescue to the symbolism of darkness; the ability of someone who has been there before, so to speak, and thus knows how to reach out. Thus, Rita’s candid words indicate a deep understanding of the survival aspect of WST. Apparently, the process that Rita describes can be conceptualized through the term “rescue fantasy”. Originating in the field of psychotherapy, it is intended to warn therapists against the danger of overly identifying with their patients. In contrast, it seems that the survival aspect embodied in Rita’s explanation legitimizes the STSM action, and it creates a deep understanding of how much mentoring, as a critical practice, incorporates active involvement on the part of the mentee.

Emily demonstrates:

**Emily**: When we go out into the street with the mobile [support unit], the women we make contact with there are scared and closed and confused… and I understand exactly at the deepest level what she is going through, and I can just manage to say the most correct word there is. And as I said, this is worth a life and this can make a change… Really, I don’t doubt any other role, in any system yes, but in some cases, it’s the key. Simple as that.

Thus, the deep understanding of those “who were there” in the past can be translated into choosing the right words and actions with the potential to save lives. However, it seems that this is not intended as a grandiose move:

**Anat:** That’s what I’m here for, to get them out of her place, I feel chills now... to be a part, to save people... is [this] taking the role of God? No, to be a part, to do what can be done, what I can do... to beat the demon, … when they tell me that she is in [the shelter] and let’s meet her and everything is fine with her, I get very excited every time. I ask how she is every holiday, she sends me such letters that I can cry, she writes to me that “it’s all thanks to you” and I wrote to her back, “no, it’s not just me, it’s fifty percent of you, it’s not just me, it’s us, together”.

Indeed, mentoring is not about taking a position of condescension toward WST; the excitement on learning of an operation that will become a shared effort, directed toward saving lives, once again underlines the narratives those that were once “there”— now rehabilitated and returning to lend a hand, quite possibly the same “hand” that they had needed themselves so long ago.

### 3.5. Growth: Mentoring as the Transformation of Pain into Growth

The research participants described their perceptions of mentoring as being a practice based on a shared destiny. Following this line of thought, they described the ways in which helping women in a situation with which they could intimately identify prompted recollections of—at times—overwhelmingly painful past experiences, but that it also created an opportunity for personal growth and for continuing their personal work on themselves. Thus, all the participants shared that being a mentor for WST brought back painful memories. Below are some examples:

**Alma:** Many of the memories came in flashbacks while working... sometimes it’s a smell, entering a discreet apartment [of women in prostitution] and smelling the candles... light... the color of a sofa... [then] when you’re in a flashback, dreams/nightmares come. After that, I have dreams at night... that I am in prostitution and all my shoes are full of money and I wake up in a panic, not understanding I was or wasn’t...

The daily encounter with WST presents the mentors with sensory triggers of the traumatic experiences that they themselves had passed through, sometimes in the form of nightmares and dreams, in other cases through other physical manifestations:

**Anat:** Prostitution has a smell, and the girls who came back [to the shelter] at five in the morning came with the smell of prostitution...and I would come home after a shift without sleeping, and [would spend] many hours scrubbing my body as if it was me working in prostitution and not them...I realized that I was experiencing, as they say, a secondary trauma and it was for me like [experiencing it] all over again.

The triggers also prompted specific memories:

**Masha**: All kinds of memories [stirred], memories that I also had when I was a teenager, of gang rape... I had been sexually harassed from the age of seven—that I can remember. I was gang-raped at the age of fourteen there were four of them who raped me, and all kinds of stories started coming back to me … [and] I realized that I can’t … I can’t smell a man next to me, I’m on the bus [and start] acting like a weirdo...

Talia and Masha here describe how smells and other triggers can trigger painful memories and even lead to repetitive behaviors such as compulsively scrubbing the body in an attempt to “cleanse” oneself. The difficult memories of severe injuries and rape emerge in the form of day-to-day difficulties. The examples that the mentors cited are consistent with what is known to women in the sex trade and their complex experience of dealing with early traumas. However, alongside the painful encounter with past experiences—some directly related to engaging in ST, others to the events that led them to engage in ST—the mentors describe how mentoring gives them a platform to transform these experiences into personal and professional growth.

**Talia:** Keeps me as a part of the field, because I also take care of myself in the process... you take care of yourself, take care of your patterns in the process… this is always what keeps me, apart from God… being in the field of mentoring keeps me safe.

Being a mentor for WST creates a space for self-compassion, for developing an awareness of destructive patterns, and to feel that they are taking ownership of a process through which they can preserve themselves, thus softening the confrontation with past traumas:

**Emily:** I was afraid of the big world… and I didn’t really know other sides of me and in my opinion to stay in the field of therapy (mentoring) it was exactly that, that is, being there with a mentee who is going through a process very similar to mine. Being with her pain is like exploring my pain together with her, even more deeply.

In this way, mentoring, as a space of joint investigation, changes the power relations between the participants, mediating the pain originating from past experiences. Instead of it taking them over, they “dismantle” it.

**Katya:** I feel grateful for where I am today really, like, with all the difficulty I say, wow how grateful that I’m really doing this… well maybe it’s not always every moment, but I’m like, I catch myself when I’m having a hard time, when I’m sad, then I suddenly see and understand, like, I look at what I have today, and what I didn’t have then, you understand?

The role of the mentor, even when it leads to dealing with potentially overwhelming moments of sadness and pain at times, gives meaning to the personal process that they are going through, and to the gap between their past and the present. It seems that unlike dealing with traumatic experiences, which is characterized by a lack of control and helplessness, the personal and professional process that is embodied in the training of the mentors, together with their role, creates a space for personal and professional growth.

## 4. Discussion

The findings of the study detail how the concept of the wounded healer is embodied in the role of the STSM providing support for WST in a way that shatters the helper-helped dichotomy, replacing this with a parallel movement between wounded experiences from the past and providing assistance to others in the present—translated by the mentors into a path of self-help and ongoing personal-psychological work.

The findings point to the importance and centrality of the mentor-mentee relationship, both as a main component of the mentorship’s contribution to the rehabilitation of WST and as a relationship in which the figure of the wounded healer is embodied (Figure 1). These observations are supported by studies that have explored the meaning of peer support from the perspective of WST, highlighting how peer support enabled them to dream of leaving the sex trade themselves by making practical advice available; raised their self-esteem; and created an opportunity and a space for them to share experiences that they would not have found elsewhere [11,14,43].

To understand and discuss the broad meaning of the alliance built into the STSM mentoring framework, its four components (mutual identification and shared destiny, parenting, hope, and saving lives), and their essential link to the combination of pain and personal growth which characterizes the experience of STSMs themselves, we draw from critical theories of mentoring in this section, making specific reference to the weight they give to the therapeutic alliance. These elements, and their combination, are consistent with several elements of a critical theory of mentoring: striving for equality, social empathy, recognition, and solidarity (Figure 1).

### 4.1. Equality

The element of equality refers to a reduction of the power relations that are embodied in expert/client relations with the goal of challenging oppressive social-gender structures embedded in the experience of injury and trauma. The ultimate objective is the creation of an experience of partnership [44,45] running contrary to the presumption in traditional mentoring of the mentor as an exclusive “expert” [28]. According to Laura Brown [46], egalitarian relations are an integral aspect of critical practice, working to reduce the mystique that preserves the power of the possessor of knowledge and expertise—and the patient as an entity lacking in this knowledge. Thus, equality is reflected in the findings, first through mutual identification and shared destiny as a moral starting point [47]. The ability to look “at eye level” [21] serves as a basis for active action when it comes to the experience of re-parenting and when it comes to the aspiration to create hope and save lives.

Reinforcement of the importance of equality is also reflected in the limited body of research about mentoring and WST. According to one study, WST reported that when the mentoring was based on the experiential knowledge that came from the mentees sharing their own stories, they perceived the mentoring as more meaningful [48]. In other studies, the contribution of establishing relationships at “eye level” and the hope for change were emphasized [11]; a sensitive support relationship based on the erasure of hierarchy, which contributed to establishing a relationship between WST and the aid organization, was explored [43]; and cooperation between mentor and mentee was found to be a factor with a positive effect on the perception of WST regarding its contribution to their rehabilitation [14].

### 4.2. Critical Empathy

The traditional concept of mentoring and helping relationships presents the importance of the concept of the wounded healer as enabling the building of an empathic bridge to the emotional experience of the patient [16]. The findings of the current study, however, suggest a broader perspective than that of the emotional world, referring to the embodiment of the wounded healer in mentoring from a critical perspective. Considering the elements of parenting as a curative experience, as well as the creation of hope and a successful life, the ability of the STSM to make use of the bridge between their pain and professional growth and translate it into actions that go beyond the emotional dimension of helping relationships is critical. Examples of such operations are protecting WST from violence, preparing a sweet dish for them against the rules of the hostel, or holding onto the knowledge that WST have dreams for a better future which are sometimes repressed.

Guided by the lens of a critical theory of mentoring, the actions of STSMs expand into social-critical empathy—that is, into a unique form of attention given to social suffering, and to those who suffer from it [49]. This type of process is anchored in intersubjective theory [50], confirming the way in which the STSMs can base their relationship with WST on a notion of intersubjective communication in which they also position themselves as a subject who has undergone social pain, using this social pain as a compass for exploring the social conditions that shape the suffering of WST.

Thus, it is possible to differentiate empathy from critical empathy in the engagement as mentors of STSMs with WST through the distinction between empathy as an intrapsychic process and empathy as a social process. When it comes to empathy as an intrapsychic process, the therapist (object) gives meaning to the negative feelings of the subject (patient/mentee). Critical empathy expands this position into an inter-subjective process in which the mentor gives moral meaning to feelings and suffering originating in social conditions, an action that is also political [49]. Indeed, the importance of critical empathy is supported by intersubjective theory, which deals with the “psychology of two people”. This theory emphasizes the partnership and relationship between a person and another person as two subjects and sees the development of a person as arising from the interpersonal relationships in which he or she is involved [50].

Thus, showing empathy for WST is different from showing empathy for the victims of a social disaster. In the first example, women are seen as responsible for their situation, but in the second example the underlying factor is the force of nature. Therefore, the effort necessary in expressing empathy for WST is fundamentally political because it opposes the labeling and pathologizing of the sex trade and those engaged in it, instead focusing on the social circumstances of this suffering. The critical social empathy of STSMs, informed in their work by the concept of the wounded healer and by positioning themselves in the role of mentors as peers of the helped, derives from a horizontal model of empathy [49]. This differs from the hierarchical model, which is anchored by an authoritative relationship between the professional and the helped.

### 4.3. Recognition

Through the lens of critical mentoring theory, the issue of recognition is linked to the stigmatization that STSMs experienced in their history as WST, which is associated with ostracism and social infamy [12,51]. In this sense, while stigma is a political mechanism that feeds into a lack of recognition [52], the social stigma of the sex trade leaves irreversible damage on a person’s self-identity and basic dignity. This process is accompanied by feelings of shame, concealment, and self-hatred, which endure even after a woman leaves the sex trade [51]. Thus, the right of WST to dream of a new life is blocked by both social and psychological barriers created by the internalization of this self-loathing. Through this narrative of self-loathing, the WST experience a form of reduction into “flat” and two-dimensional characters in the story seen only through the lens of “prostitutes” and subjected to the demeaning observation that “once a whore, always a whore” [53].

Thus, social stigma has consequences for the STSM; when they choose to support WST, they relive the experience of exclusion and shame, as described in the findings section of this paper. That is, their choice of this role may have, among other things, psychological consequences—both in relation to the re-awakening of past personal trauma and to the trauma originating in the social stigma.

If so, taking a critical point of view, the four different components of the actions of STSMs converge together in a process aimed at recognizing the right of WST to dream, their right to rehabilitation, and their right to reconnect to their wounded body and the traumatic mental experiences embodied within—and to connect to positive areas of hope, development, and personal growth, as reflected in the mentorship relationships themselves. In this sense, the deep understanding that STSMs bring to the relationship about the self-contempt and the social death of sex workers [54] that characterize the meeting point between social processes, and between psychological mechanisms of hatred and self-loathing, can be seen as originating from their positioning as the wounded healer. The act of mentoring takes on the meaning of not only saving physical life, but also of trying to save of the psychological life of WST. In this sense, it is about combining the feminist aspects of critical mentoring with barrier transcendence theory [28,33].

### 4.4. Solidarity

Solidarity, an essential element in the critical theory of mentoring, borrows its strength from the idea of sisterhood in feminism [44]. The role perception of the STSM, and the role played by the wounded healer in the mentor-mentee relationship, is consistent with the concept of solidarity. This emanates from the STSMs’ capacity to see WST as they see themselves, and as belonging to the same sex worker critical community of care [28,55]—a belonging that translates into active responses by STSMs in protecting the WST as a form of active advocacy [56,57,58]. This can be seen as a possible response to the claim that WST are denied solidarity in the mainstream discourse on the sex trade when solidarity as an aspect of the STSM role translates into de-stigmatization and de-criminalization [59].

Thus, the solidarity expressed by STSMs gains value—more so if considered alongside the claim of a lack of solidarity between WST [60] due to the presence of an internal hierarchy among them and due to their diminished social status. Sex workers experience relationships that can be used as weapons against them, making it hard for them to vest their trust in others. Therefore, as Shemai [58] pointed out, the relationship is ultimately about solidarity in a space of conflict. In this sense, the STSM point of view, which considers the enormous vulnerability of WST, and the importance of their personal exposure as mentors serve as a basis for building any experience of solidarity, and in the aspired transition from anonymity to recognizing them as subjects.

## 5. Conclusions

The current study develops the relatively limited existing body of knowledge about mentoring as a unique form of peer support for WST survivors and as a type of rehabilitation intervention for WST—an intervention that offers an alternative to the traditional concept of the expert/client relationship. Relating to STSMs as a specific population group, the study holds up the unique semi-professional position that the mentors occupy. Through this, they can hold professional positions and perspectives, but at the same time can diminish the elements of distance and/or suspicion that are often an aspect of the working relationship between WST and welfare professionals. For this reason, we recommend the adoption and further development of mentoring in rehabilitation work with WST—both as an intervention that stands on its own and as a regulated and supervised means of building trust and closer communication between the WST—by welfare organizations and by professional therapists.

Referring to the research field about mentoring and specifically critical mentoring, one can learn about the ways in which STSMs can offer an expansion of the qualities and meanings that are vested in the concept of the wounded healer. Indeed, it is recommended that mentoring programs for excluded and marginalized populations adopt the broad understanding of mentoring that those who have “been there” in the sex trade can offer—the imperative, deriving from the critical point of view, to take an active stance of assistance. Following this approach, mentoring becomes an act of solidarity, equality, critical empathy, and recognition, crucial in work with population groups experiencing forms of social oppression.

This means that mentoring, as an intervention method, moves from the emotional aspect of the experience into shaping curative experiences of parenting, support, mutual identification and shared destiny, hope, and saving lives, in the physical and philosophical sense. Moreover, the findings highlight how the role of mentoring serves STSMs themselves. It creates a space for reliving their own experiences again and again and meeting once more with the pain, although in a form that can be described as controlled exposure (contrary to the lack of control that they had experienced in the past). This provides the foundation for a process of personal and professional growth. Therefore, we recommended that careful thought be given to developing mentoring as a practice that supports personal care for women leaving and recovering from the ST, while taking into consideration the complexities that the role can create for the mentors given the triggers that are inevitable in their encounters with WST.

Thus, alongside the significant contribution of mentoring as a practice of rehabilitation and recovery, both for WST and for STSMs, the challenges that accompany this role should be considered. It must be remembered that mentoring is defined as a semi-professional position; the STSMs are part of a profession that will protect them, both in relation to their employment conditions and in relation to professional aspects of the position such as training, emotional support, occupational continuity, and recognition of their unique knowledge. STSMs play an emotional-supportive role in that they serve as containers for the severe injuries and traumas of WST. As shown in the results section, this role poses the risk of re-awakening past trauma and imposes a tremendous psychological burden. To this end, training and professional supervision must be guaranteed and regulated as an inherent part of STSM working conditions. Finally, some of the STSMs described the real dangers that they are exposed to on the street during the outreach process with WST. These dangers must be considered to guarantee their physical and mental safety when performing such a complex role.

This study has a number of limitations. First, the research is based on a series of interviews with a small and unique population group of experienced mentors. As such, it is limited by broader challenges in the research field on WST. Future research on the topic should collect additional data about mentoring from (amongst others) young mentors and from different points in time in the rehabilitation process. Secondly, this study does not consider the perspectives of WST themselves vis-à-vis their experiences of the mentoring relationship. A complementary point of view is required to fully conceptualize the mentoring relationship. Thirdly, because it is based on interviews the current research does not allow for examination of the process in the here and now. We suggest that future research engage with this issue by way of ethnographic studies underwritten by the requirement to build trust with WST and the mentors who support them and through close examination of the rehabilitation processes that are supported by mentoring work. Finally, this study draws from a relatively homogeneous sample in that it does not incorporate the voices of other people working in the sex trade, such as women of varying religions (which is particularly relevant for the setting of the study) and the LGBT community.

## Figures and Tables

**Figure 1 ijerph-20-04089-f001:**
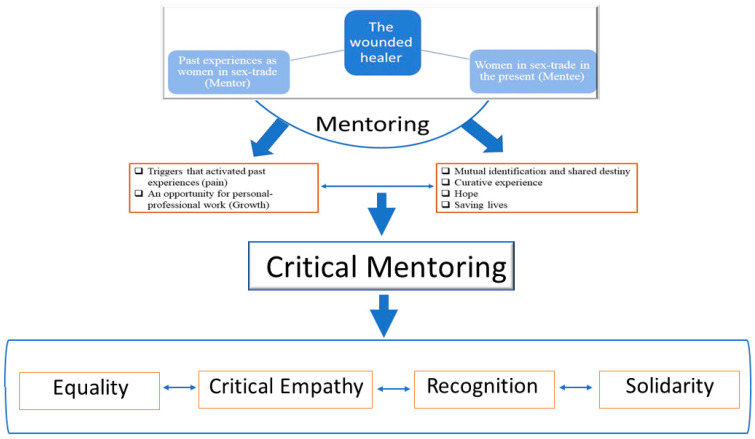
The wounded healer from critical mentoring perspective.

## Data Availability

Data are contained within the article. The full dataset is not publicly available, due to considerations.

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
