# Peer review of "It’s Not Just Me, It’s Us, Together: The Embodied of the Wounded Healer in the Role of Sex Trade Survival Mentors—A Critical Mentoring Perspective"

_ijerph, 2023, doi:10.3390/ijerph20054089_

Round 1

Reviewer 1 Report

The present study utilizes a critical-feminist qualitative approach to understanding the experiences of mentors assisting women in and leaving the sex trade. The study explores how these mentors, themselves survivors of the sex trade, interact with their mentees, make meaning from these experiences, and the emotional toll these experiences have on them. The study provides useful data about a very understudied topic, helping women leave the sex trade, and a very understudied group, survivor-mentors ("wounded healers"). Overall this study provides a unique and important contribution to understanding a complex social issue from the perspective of marginalized individuals.

Consider addressing the following concerns:

Error on page 11, under Rita's quote, the authors say "Emily ties the element of...", should be Rita instead of Emily

Include figure 1 on page 13

The first full paragraph on page 15 (beginning "Thus, the difference between empathy and critical empathy...") is a bit hard to understand and would benefit from clearer wording.

Please consider rewording this portion of the sentence on page 15: "the social stigma of the sex trade expands the understanding of a process in which the negative image (stigma) converts the damage is causes into irreversible damage to the person's self-identity and basic dignity" as it's hard to understand. 

There is a strong focus on the experiences of WST in 4.3 Recognition in the discussion section, rather than focusing on the experiences of STSM. It may be worth considering refocusing this section on the mentor's experiences given the study's aims. 

Author Response

Reviewer 1

See page

Error on page 11, under Rita's quote, the authors say "Emily ties the element of...", should be Rita instead of Emily

We changed the name Emily to Rita.

13

Include figure 1 on page 13

We have included a reference to Figure 1 on this page

16

The first full paragraph on page 15 (beginning "Thus, the difference between empathy and critical empathy...") is a bit hard to understand and would benefit from clearer wording.

We rewrite the sentences in question

17

Please consider rewording this portion of the sentence on page 15: "the social stigma of the sex trade expands the understanding of a process in which the negative image (stigma) converts the damage is causes into irreversible damage to the person's self-identity and basic dignity" as it's hard to understand. 

We rewrite the sentences in question

18

There is a strong focus on the experiences of WST in 4.3 Recognition in the discussion section, rather than focusing on the experiences of STSM. It may be worth considering refocusing this section on the mentor's experiences given the study's aims. 

We have included in this part explanations about the STSM experience and its connection to their history as WST

17, 20

We would like to thank the reviewer for her\his\their feedback and contribution to improving the manuscript

Reviewer 2 Report

General

The authors write about a scarcely covered topic – peer mentors in the sex trade world. Their study provides a great contribution to extant literature, despite their small sample. The findings are very interesting and well organized. Their methodology is nicely detailed. In terms of content, I think it would be a great fit to the special issue of the journal.

The manuscript would benefit from additional language editing. Prepositions are often absent in sentences throughout the manuscript, some sentences are grammatically incorrect, and there are a few typos throughout the manuscript. Some sentences are very long (6-7 lines) and difficult to follow.

Abstract

 Typo line 3 – should be “mentors” (plural)

The authors use the term “sex trade,” which is an umbrella term; I believe what the authors refer to more specifically is street-based sex trade, because all their references to vulnerability, physical and mental health problems may be unique to women on the street and not to all types of sex-trade involved people (for example, high-end escorts).

Background

Overall, this section lacks many citations throughout. I would encourage the authors to review this section and add citations where they currently have factual statements that are not supported by specific literature.

Typo p. 2- missing “are” after “they” in second paragraph, line 7 from the end of the paragraph

In that same paragraph, authors are citing “a single case study” which “points to the importance of the therapeutic alliance” – but that’s a 2012 study which has been cited 18 times since! It would be better to check for additional case studies to demonstrate authors’ point.

P. 2 – I would suggest that the authors define what therapeutic alliance is, before they write about its applicability to WST. Similarly, they should distinguish (or describe) the specific mentors they refer to: Not all mentors are peer support coaches/mentors; in other words, women, including WST, can have mentors who have “not been there.”

The authors do provide a citation for the “wounded healer,” but in the context of the manuscript, it’s best to define it for the reader, in light of the importance of this concept. It is described and defined in the following section, but that should be in the first mention of the concept. In other words, the authors should rearrange the content of their introduction sections. At the very least, they should provide a brief definition of the term when first mentioned on p. 2.

The last sentence on page 2 is very long, cumbersome, and partially grammatically incorrect (missing a preposition in the end). The next sentence (beginning of p. 3) is grammatically incorrect and unclear.

The authors write on p. 3, “As part of their professional training, practitioners are trained to shatter the therapist-patient dichotomy, making intelligent and controlled use of these wounds and painful personal experiences in building an empathetic bridge toward their patients.” From a psychological perspective, this statement is unclear: It is certainly not the common paradigm or professional training for therapists to “shatter the therapist-patient dichotomy; more likely it is the opposite. If the authors refer in this sentence to peer support (and not “practitioners”), they should clarify it.

p. 3 – Critical mentoring

The authors write, “mentoring, as a unique form of peer support, is an accepted practice in the helping professions.” Semantically speaking, I would argue that peer support is a unique form of mentoring (one in which the mentor has similar lived experience as their mentee), and not vice versa as the authors write.

p. 4 – I would recommend rewriting the sub-section on critical mentoring. It is not very clear, and it serves as an important basis for the Discussion.

p. 4 – while I see the value of applying a feminist perspective to mentoring, the theory is introduced rather abruptly and feels disconnected from the rest of the paragraph. This is even more true of the Barrier transcendence theory (p. 5). My general impression is that the authors collected multiple theories and threw them in without true relevance to their own study, which focused, as they wrote, on examining the ways in which the wounded healer principle applied to the rehabilitation of WST. The authors should present a better integration of their theoretical framework and show the relevance of the theories to their research questions.

p. 5 - The research questions are phrased in a truly cumbersome way. I had to read several times to understand that the first one looked at the contribution of STSM to the rehabilitation of WST from the mentor’s perspective, and the second one was about how being a mentor contributes to the mentor’s self-perception.

Methodology

Especially since there are so few participants, it would be nice to have a table providing specific information for each one – marital status, experience as mentors, years in ST, years out of the ST, as well as age and other demographics if available (substance use, number of children, etc.).

p. 6 - It would be great if the authors could list a few more questions on the interview guide, other than just the opening question, especially since they wrote it was a semi-structured interview.

The authors write a very vague sentence regarding the study’s reliability – what kind of triangulation and what comparative data did the researchers possess? Please elaborate on this important point.

p. 7 - Again – for understanding reliability, the authors write that they received feedback at three academic conferences, but do not detail the nature of the feedback or how it was incorporated into their study.

Were participants compensated for their participation in the research?

Did participants pick their own pseudonym or were those selected for them by the researchers?

Results

The Results section is generally well written: The quotes are nicely introduced before and discussed afterwards. I found the quotes to be very illuminating. Some of the longer ones could probably be trimmed a little for better clarity.

My issue is with the headings of two of the subsections (themes). In the first sub-section, the name of the theme “reciprocity” isn’t quite appropriate to the quotes in that section. With the exception of one quote (Talia’s on p. 8), the other quotes are more about similarities and differences (as the subtheme relates) between the mentors and mentees and not about the exchange of benefits between them. I think the authors should revise the name of this theme to exclude “reciprocity.” It is incorrect and confusing.

Similarly, the second theme “Corrective experience: being in the position of a parent” is about being in a position of a parent but not necessarily about corrective experiences (which are a form of therapy). The interviewees do not mention that they are acting to replace the mentee’s parents, or providing a new experience, only that they are acting like a parent. I suggest revising the heading to exclude “corrective experience.”

p. 11 – In the last sentence before section 3.5 – the authors write “…those that were once “there” – now rescued and returning…” . “Rescued” is quite a loaded term; I think it’s better to replace “rescued” with “rehabilitated,” especially since we do not know anything about the mentors’ recovery process.

Discussion

On p. 13 there’s a note “Add figure 1,” but this figure is no where to be found. I therefore could not relate to it.

Top of p. 14 – shouldn’t this be “we draw” instead of “I draw”, as there are two authors?

P. 14 – the sentence that begins with “The ability to look…. And save lives” is incredibly long, cumbersome, and unclear.

p. 14, last paragraph before 4.2 – the authors do not engage with the few of the studies of mentoring and WST. While those are few and far between, there is more than one they could relate to (e.g. Thorlby, 2015; Preble et al., 2016; Woodman, 2000).

p. 15 - The authors bring up yet another theory, the intersubjective theory, which may well be relevant, but should be better introduced and integrated with their findings.

Large parts of the discussion seemed to be dedicated to the experiences of WST, rather to that of their mentors. While it is of course relevant and ties in with the overall narrative, I thought the findings could be somewhat better discussed. For example, I didn’t see any discussion of the fact that the STSM seemed to be paying a psychological price for their mentoring, in terms of having flashbacks or obsessive compulsive behavior after mentoring sessions. I don’t know if this is common for all wounded healers, but it certainly merits mentioning in the discussion. Could there be a way to alleviate these “side effects” of mentoring? The conclusion relates to this issue, but from a slightly odd perspective – as if living and reliving the pain is a good experience for the mentors (p. 17). Either way, I was hoping to see a more thorough discussion of this point in the body of the Discussion.

Author Response

Reviewer 2

Abstract

See page

 Typo line 3 – should be “mentors” (plural)

We have made changes in accordance with this recommendation

1

The authors use the term “sex trade,” which is an umbrella term; I believe what the authors refer to more specifically is street-based sex trade, because all their references to vulnerability, physical and mental health problems may be unique to women on the street and not to all types of sex-trade involved people (for example, high-end escorts).

We added a clarification note at the beginning of the article and made corrections throughout the article.

2

Background

Typo p. 2- missing “are” after “they” in second paragraph, line 7 from the end of the paragraph

In that same paragraph, authors are citing “a single case study” which “points to the importance of the therapeutic alliance” – but that’s a 2012 study which has been cited 18 times since! It would be better to check for additional case studies to demonstrate authors’ point.

We have made changes in accordance with this recommendation.

2

2

P. 2 – I would suggest that the authors define what therapeutic alliance is, before they write about its applicability to WST.

Similarly, they should distinguish (or describe) the specific mentors they refer to: Not all mentors are peer support coaches/mentors; in other words, women, including WST, can have mentors who have “not been there.”

We have included clarification about the concept in the text

We added a clarification note

3

3

The authors do provide a citation for the “wounded healer,” but in the context of the manuscript, it’s best to define it for the reader, in light of the importance of this concept. It is described and defined in the following section, but that should be in the first mention of the concept. In other words, the authors should rearrange the content of their introduction sections. At the very least, they should provide a brief definition of the term when first mentioned on p. 2.

We have made changes in accordance with this recommendation.

2, 4

The last sentence on page 2 is very long, cumbersome, and partially grammatically incorrect (missing a preposition in the end). The next sentence (beginning of p. 3) is grammatically incorrect and unclear.

We have made changes in accordance with this recommendation.

3

The authors write on p. 3, “As part of their professional training, practitioners are trained to shatter the therapist-patient dichotomy, making intelligent and controlled use of these wounds and painful personal experiences in building an empathetic bridge toward their patients.” From a psychological perspective, this statement is unclear: It is certainly not the common paradigm or professional training for therapists to “shatter the therapist-patient dichotomy; more likely it is the opposite. If the authors refer in this sentence to peer support (and not “practitioners”), they should clarify it.

We have made changes in accordance with this recommendation.

4

Critical mentoring

The authors write, “mentoring, as a unique form of peer support, is an accepted practice in the helping professions.” Semantically speaking, I would argue that peer support is a unique form of mentoring (one in which the mentor has similar lived experience as their mentee), and not vice versa as the authors write.

We have made changes in accordance with this recommendation.

4

p. 4 – I would recommend rewriting the sub-section on critical mentoring. It is not very clear, and it serves as an important basis for the Discussion.

We have made changes in accordance with this recommendation.

4, 5, 6

p. 4 – while I see the value of applying a feminist perspective to mentoring, the theory is introduced rather abruptly and feels disconnected from the rest of the paragraph. This is even more true of the Barrier transcendence theory (p. 5).

My general impression is that the authors collected multiple theories and threw them in without true relevance to their own study, which focused, as they wrote, on examining the ways in which the wounded healer principle applied to the rehabilitation of WST. The authors should present a better integration of their theoretical framework and show the relevance of the theories to their research questions.

We have made changes in accordance with this recommendation.

4, 5, 6

p. 5 - The research questions are phrased in a truly cumbersome way. I had to read several times to understand that the first one looked at the contribution of STSM to the rehabilitation of WST from the mentor’s perspective, and the second one was about how being a mentor contributes to the mentor’s self-perception.

We have made changes in accordance with this recommendation.

6

Methodology

Especially since there are so few participants, it would be nice to have a table providing specific information for each one – marital status, experience as mentors, years in ST, years out of the ST, as well as age and other demographics if available (substance use, number of children, etc.).

We have included in the text (2.5. Ethical considerations) an explanation of why this information remains confidential

6, 7

p. 6 - It would be great if the authors could list a few more questions on the interview guide, other than just the opening question, especially since they wrote it was a semi-structured interview.

We incorporated examples from the interview guide within the section of the research tool

7

The authors write a very vague sentence regarding the study’s reliability – what kind of triangulation and what comparative data did the researchers possess? Please elaborate on this important point. p. 7 - Again – for understanding reliability, the authors write that they received feedback at three academic conferences, but do not detail the nature of the feedback or how it was incorporated into their study.

We have made changes in accordance with this recommendation.

8

Were participants compensated for their participation in the research?

We have included an explanation regarding this issue in the text

8

Did participants pick their own pseudonym or were those selected for them by the researchers?

We have included an explanation regarding this issue in the text

8

Results

My issue is with the headings of two of the subsections (themes). In the first sub-section, the name of the theme “reciprocity” isn’t quite appropriate to the quotes in that section. With the exception of one quote (Talia’s on p. 8), the other quotes are more about similarities and differences (as the subtheme relates) between the mentors and mentees and not about the exchange of benefits between them. I think the authors should revise the name of this theme to exclude “reciprocity.” It is incorrect and confusing.

We have made changes in accordance with this recommendation.

9, 10, 16, 20

Similarly, the second theme “Corrective experience: being in the position of a parent” is about being in a position of a parent but not necessarily about corrective experiences (which are a form of therapy). The interviewees do not mention that they are acting to replace the mentee’s parents, or providing a new experience, only that they are acting like a parent. I suggest revising the heading to exclude “corrective experience.”

We have made changes in accordance with this recommendation.

10, 17, 20

p. 11 – In the last sentence before section 3.5 – the authors write “…those that were once “there” – now rescued and returning…” . “Rescued” is quite a loaded term; I think it’s better to replace “rescued” with “rehabilitated,” especially since we do not know anything about the mentors’ recovery process.

We have made changes in accordance with this recommendation.

13

Discussion

On p. 13 there’s a note “Add figure 1,” but this figure is no where to be found. I therefore could not relate to it.

We have added one figure to the text

16

Top of p. 14 – shouldn’t this be “we draw” instead of “I draw”, as there are two authors?

We have made changes in accordance with this recommendation.

16

P. 14 – the sentence that begins with “The ability to look…. And save lives” is incredibly long, cumbersome, and unclear.

We have made changes in accordance with this recommendation.

16, 17

p. 14, last paragraph before 4.2 – the authors do not engage with the few of the studies of mentoring and WST. While those are few and far between, there is more than one they could relate to (e.g. Thorlby, 2015; Preble et al., 2016; Woodman, 2000).

We have made changes in accordance with this recommendation.

17

p. 15 - The authors bring up yet another theory, the intersubjective theory, which may well be relevant, but should be better introduced and integrated with their findings.

We have included an explanation regarding this theory in the text

18

Large parts of the discussion seemed to be dedicated to the experiences of WST, rather to that of their mentors. While it is of course relevant and ties in with the overall narrative, I thought the findings could be somewhat better discussed. For example, I didn’t see any discussion of the fact that the STSM seemed to be paying a psychological price for their mentoring, in terms of having flashbacks or obsessive compulsive behavior after mentoring sessions. I don’t know if this is common for all wounded healers, but it certainly merits mentioning in the discussion. Could there be a way to alleviate these “side effects” of mentoring? The conclusion relates to this issue, but from a slightly odd perspective – as if living and reliving the pain is a good experience for the mentors (p. 17). Either way, I was hoping to see a more thorough discussion of this point in the body of the Discussion.

We have made changes in accordance with this recommendation.

20

We would like to thank the reviewer for her\his\their feedback and contribution to improving the manuscript

Reviewer 3 Report

The women described in this study are brave, and some of them have faced difficult situations, violence and rape, drug addiction, prison, and work in the sex trade on the streets or under dire conditions. Despite these hardships, they managed to undergo a rehabilitation process and become mentors. The article makes a significant contribution by shedding light on how STSMs base their relationships with the WSTs on the notion of intersubjective communication, position themselves as subjects who have experienced social pain, and use this pain as a compass for exploring the social conditions that shape the suffering of WSTs. In this way, the article succeeds in meeting its aim of examining how the past experiences of STSMs support the embodiment of the wounded healer approach to mentoring. To ensure the manuscript is suitable for publication, I suggest that the authors make several changes and clarifications in it, as I describe below. 

Introduction

  • Since the research focuses on how the notion of the wounded healer is embodied in the STSM role, the authors should consider structuring the introduction differently. Beginning with “the wounded healer” or “critical mentoring” and only then introducing “women in the sex trade” will add another aspect of critical feminist methodology (focusing on the phenomenon being studied and what we can learn from STSMs). Moreover, this step will make the introduction clearer to the reader.
  • The population described in the study is among many groups within the broad field of the sex trade. Therefore, I suggest the authors refer to this point and emphasize that this a particular group and not all those engaged in the sex industry are addressed in this manuscript. I suggest clarifying this point to prevent further general stigmatization of people who engage in the sex trade. One of the manifestations of this issue in the manuscript is the use of different terms to refer to the population, for example, “sex worker” (p. 8), “patient” (p. 9), and “sex-trade survivor” (p. 9). Each term has a different meaning in sex trade theory, practice, therapy, and activism. I suggest the authors decide on the term that best describes the particular population and use it consistently, as far as possible, throughout the article. It would be best to add a brief explanation to clarify the motivation for using the chosen term.
  • On pages 1 and 2, the authors use the phrase “these women”. Please consider rephrasing this; something about this expression emphasizes women’s otherness.

Method

  • In the article, there is not much reference to the personal rehabilitation processes of the STSMs. I suggest the authors add something about this in the participants’ descriptions if this information is available. 
  • I suggest adding context regarding the organisation and the space in which STSMs operate, a definition of their role, an explanation of how much freedom of action they have, and a description of the supervision and support they receive if this information is available. 
  • On p. 7, the authors write: “We also received feedback at three academic conferences, one a dedicated conference on the topic of WST”Please say something about the feedback received and how it supports your findings.

Ethical considerations (p. 7)

  • The authors state they used the ethics of care approach and a critical feminist approach. Did issues related to emotional flooding arise following the interviews? If so, how did the authors ensure the participants’ safety?

Results

  • I suggest the authors reconsider their use of the phrase “corrective experience” in the second theme. Perhaps “curative experience” would better express this point. 
  • On p. 9, the authors write: “Anat calls the sex workers whom she supports ‘girls’. It seems that they wanted to say something about this. What is significant here in the context of the theme? What is the importance of using the term “sex work” and not the term WST? These points should be clarified and sharpened. 
  • “Saving lives” should be discussed in conjunction with the issue of the rescue fantasy that exists among therapists, especially in the context of the sex trade. I want to draw the authors’ attention to the fact that Rita’s second quote (p. 11) has no reference, and precisely here lies the alternative to the rescue fantasy: “she can stay there in the dark with her and hold her hand and not run away from her”.

Discussion

There is not enough discussion of the challenges within the mentoring processes in the context of the wounded healer. I suggest that the authors integrate the challenges within the existing discussion so that their recommendation for a practice that requires sensitive planning will be clearer. 

  • p. 14, should it be “we” instead of “I”?

Author Response

Reviewer 3

Introduction

See page

Since the research focuses on how the notion of the wounded healer is embodied in the STSM role, the authors should consider structuring the introduction differently. Beginning with “the wounded healer” or “critical mentoring” and only then introducing “women in the sex trade” will add another aspect of critical feminist methodology (focusing on the phenomenon being studied and what we can learn from STSMs). Moreover, this step will make the introduction clearer to the reader.

We added a reference to this feedback at the beginning of the introduction

1, 2

The population described in the study is among many groups within the broad field of the sex trade. Therefore, I suggest the authors refer to this point and emphasize that this a particular group and not all those engaged in the sex industry are addressed in this manuscript. I suggest clarifying this point to prevent further general stigmatization of people who engage in the sex trade. One of the manifestations of this issue in the manuscript is the use of different terms to refer to the population, for example, “sex worker” (p. 8), “patient” (p. 9), and “sex-trade survivor” (p. 9). Each term has a different meaning in sex trade theory, practice, therapy, and activism. I suggest the authors decide on the term that best describes the particular population and use it consistently, as far as possible, throughout the article. It would be best to add a brief explanation to clarify the motivation for using the chosen term.

We added a clarification note at the beginning of the article and made corrections throughout the article.

2, 3, 10, 14, 18, 19

On pages 1 and 2, the authors use the phrase “these women”. Please consider rephrasing this; something about this expression emphasizes women’s otherness.

We have made changes in accordance with this recommendation

1, 2

Method

In the article, there is not much reference to the personal rehabilitation processes of the STSMs. I suggest the authors add something about this in the participants’ descriptions if this information is available. 

We addressed this issue in accordance with the recommendations

6

I suggest adding context regarding the organisation and the space in which STSMs operate, a definition of their role, an explanation of how much freedom of action they have, and a description of the supervision and support they receive if this information is available. 

We addressed this issue in accordance with the recommendations

6,7

On p. 7, the authors write: “We also received feedback at three academic conferences, one a dedicated conference on the topic of WST”. Please say something about the feedback received and how it supports your findings

We have added relevant information following this feedback

7

Ethical considerations 

The authors state they used the ethics of care approach and a critical feminist approach. Did issues related to emotional flooding arise following the interviews? If so, how did the authors ensure the participants’ safety?

We addressed this issue

8, 9

Results

I suggest the authors reconsider their use of the phrase “corrective experience” in the second theme. Perhaps “curative experience” would better express this point. 

We changed this conceptualization following the recommendation

10, 17, 20

On p. 9, the authors write: “Anat calls the sex workers whom she supports ‘girls’. It seems that they wanted to say something about this. What is significant here in the context of the theme? What is the importance of using the term “sex work” and not the term WST? These points should be clarified and sharpened. 

We expanded the explanation about the theme and changed the terminology

11

“Saving lives” should be discussed in conjunction with the issue of the rescue fantasy that exists among therapists, especially in the context of the sex trade. I want to draw the authors’ attention to the fact that Rita’s second quote (p. 11) has no reference, and precisely here lies the alternative to the rescue fantasy: “she can stay there in the dark with her and hold her hand and not run away from her”.

We expanded the explanation about the theme and changed the terminology

13

Discussion

There is not enough discussion of the challenges within the mentoring processes in the context of the wounded healer. I suggest that the authors integrate the challenges within the existing discussion so that their recommendation for a practice that requires sensitive planning will be clearer. 

We added an explanation about the challenges of STSM

20

p. 14, should it be “we” instead of “I”?

We have made changes in accordance with this recommendation

16

We would like to thank the reviewer for her\his\their feedback and contribution to improving the manuscript
